# Cost-effectiveness of mechanical thrombectomy within 6 hours of acute ischaemic stroke in China

Yuesong Pan,[1,2,3,4,5] Xueli Cai,[6] Xiaochuan Huo,[1,2,3,4] Xingquan Zhao,[1,2,3,4] Liping Liu,[1,2,3,4] Yongjun Wang,[1,2,3,4] Zhongrong Miao,[1,2,3,4] Yilong Wang[1,2,3,4]

YP and XC contributed equally.

[1]Department of Neurology, Beijing Tiantan Hospital, Capital Medical University, Beijing, China
[2]China National Clinical Research Centre for Neurological Diseases, Beijing, China
[3]Centre of Stroke, Beijing Institute for Brain Disorders, Beijing, China
[4]Beijing Key Laboratory of Translational Medicine for Cerebrovascular Disease, Beijing, China
[5]Department of Epidemiology and Health Statistics, School of Public Health, Capital Medical University, Beijing, China
[6]Department of Neurology, Lishui Hospital of Zhejiang University (the Central Hospital of Lishui), Lishui, China

**Correspondence to**
Dr. Zhongrong Miao;
zhongrongm@163.com and Dr Yilong Wang;
yilong528@gmail.com

## ABSTRACT

**Objectives** Endovascular mechanical thrombectomy is an effective but expensive therapy for acute ischaemic stroke with proximal anterior circulation occlusion. This study aimed to determine the cost-effectiveness of mechanical thrombectomy in China, which is the largest developing country.

**Design** A combination of decision tree and Markov model was developed. Outcome and cost data were derived from the published literature and claims database. The efficacy data were derived from the meta-analyses of nine trials. One-way and probabilistic sensitivity analyses were performed in order to assess the uncertainty of the results.

**Setting** Hospitals in China.

**Participants** The patients with acute ischaemic stroke caused by proximal anterior circulation occlusion within 6 hours.

**Interventions** Mechanical thrombectomy within 6 hours with intravenous tissue plasminogen activator (tPA) treatment within 4.5 hours versus intravenous tPA treatment alone.

**Outcome measures** The benefit conferred by the treatment was assessed by estimating the cost per quality-adjusted life-year (QALY) gained in the long term (30 years).

**Results** The addition of mechanical thrombectomy to intravenous tPA treatment compared with standard treatment alone yielded a lifetime gain of 0.794 QALYs at an additional cost of CNY 50 000 (US$7700), resulting in a cost of CNY 63 010 (US$9690) per QALY gained. The probabilistic sensitivity analysis indicated that mechanical thrombectomy was cost-effective in 99.9% of the simulation runs at a willingness-to-pay threshold of CNY 125 700 (US$19 300) per QALY.

**Conclusions** Mechanical thrombectomy for acute ischaemic stroke caused by proximal anterior circulation occlusion within 6 hours was cost-effective in China. The data may be used as a reference with regard to medical resources allocation for stroke treatment in low-income and middle-income countries as well as in the remote areas in the developed countries.

## BACKGROUND

Arterial recanalisation and subsequent reperfusion performed shortly after acute ischaemic stroke have demonstrated their ability to restore brain function.[1] Besides

### Strengths and limitations of this study

► A combination of decision tree and Markov model was developed in order to simulate the short-term and long-term costs and outcomes after mechanical thrombectomy for ischaemic stroke.
► The majority of the parameters used in the model, including costs, utilities and transition probabilities, were collected based on the Chinese setting, reflecting the situation in the low-income and middle-income countries.
► A limitation of the present study is that the health status and costs that resulted from additional causes other than stroke were not included in this model.
► An additional limitation is that the efficacy of the mechanical thrombectomy treatment was based on trials that were completed in high-income countries; however, these were the only data available.

intravenous recombinant tissue-type plasminogen activator (tPA) within 4.5 hours, endovascular mechanical thrombectomy is another effective reperfusion strategy, which can remove large, proximal clots rapidly and results in higher rate of reperfusion compared with intravenous tPA alone.[2] Second-generation retrievable stents can achieve higher recanalisation rates compared with first-generation devices. Although trials that have used first-generation thrombectomy devices failed to demonstrate clinical benefit compared with intravenous tPA, the five recently published clinical trials in 2015 that included second-generation devices have shown clear benefits with regard to the addition of mechanical thrombectomy to standard treatment for acute ischaemic stroke caused by proximal anterior circulation occlusions.[2–6]

Although mechanical thrombectomy with second-generation devices exhibits optimal effectiveness with acceptable safety, the main disadvantage is the high cost that requires expensive devices, highly trained proceduralists and special periprocedural support.

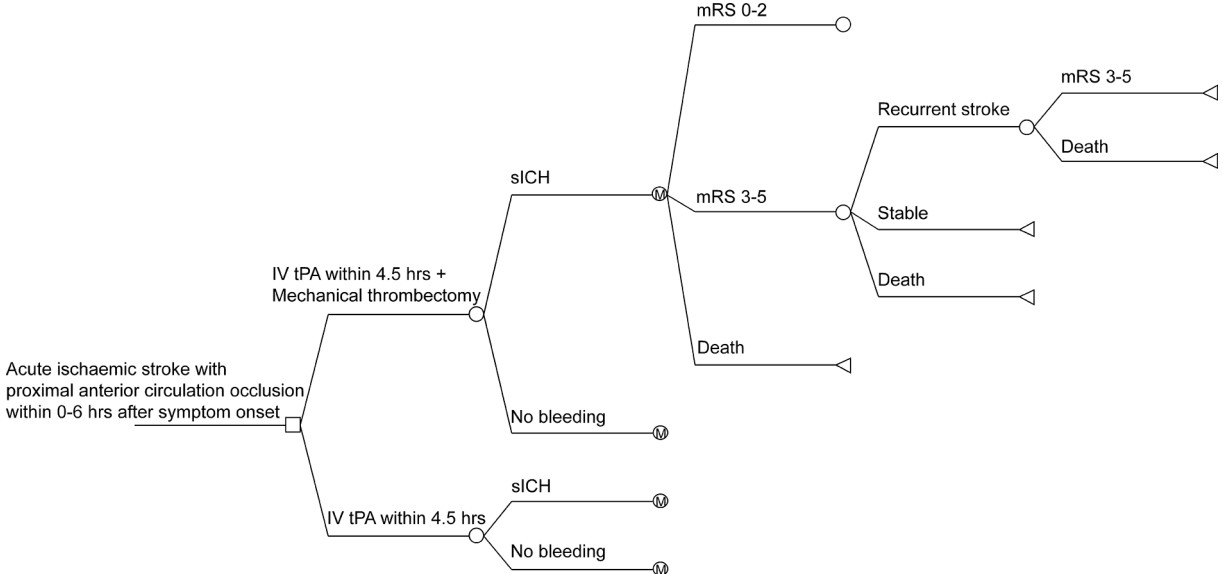

**Figure 1** Decision tree and Markov state transition model. A patient with an acute ischaemic stroke with anterior circulation occlusion entered the model at 63 years old receiving either intravenous tissue-type plasminogen activator (tPA) with or without mechanical thrombectomy and transited between health states until death or 30 years. Patients may remain in the same health state, move to a state of equal or greater disability after recurrent stroke or die. Only transition from dependent state (modified Rankin Scale (mRS) 3–5) was illustrated in the figure. IV, intravenous; M, Markov node; sICH, symptomatic intracerebral haemorrhage.

Previous studies that have examined the economic aspects of this type of therapy were conducted in the USA, UK, Sweden, Canada and Spain. These studies indicated that mechanical thrombectomy treatment for acute ischaemic stroke was cost-effective or even cost-saving in the long term.[7–15] However, all of these studies were conducted in high-income countries, and the corresponding results may not be applicable in low-income and middle-income countries such as China, where medical resources are scarce and stroke is a leading cause of death.[16] Little is known with regard to the cost-effectiveness and feasibility of the application of mechanical thrombectomy treatment in patients with acute ischaemic stroke in low-income and middle-income countries. The analysis of the economic costs involved in mechanical thrombectomy in low-income and middle-income countries is urgent. In the present study, we aimed to evaluate the cost-effectiveness of the addition of mechanical thrombectomy treatment with second-generation devices to standard care for acute ischaemic stroke in the setting of China, which is considered the largest low-income and middle-income country.

## METHODS
### Model overview
A combination of decision tree and Markov model (figure 1) was developed in order to simulate the short-term (1, 5 and 6 years) and long-term (30 years) cost-effectiveness of mechanical thrombectomy using stent retrievers within 6 hours with intravenous tPA treatment within 4.5 hours versus intravenous tPA treatment alone within 4.5 hours after onset of stroke. The base

case of the model was a cohort of 100 000 patients with acute ischaemic stroke with proximal anterior circulation occlusion (36% female), with mean age of 63 years old, being admitted to the hospital within 6 hours after onset of stroke. The baseline characteristics were the same as patients enrolled in the treated arm of the Endovascular therapy for Acute ischaemic Stroke Trial (EAST), a non-randomised interventional study that aimed to evaluate the safety and efficacy of Solitaire thrombectomy in Chinese patients with acute stroke)[17] who were admitted to the hospital within 6 hours. Among the patients admitted to the hospital within 6 hours after the onset of stroke, we assumed that 85.4% patients were admitted within 4.5 hours in both arms according to the data from the China National Stroke Registry (CNSR); a nationwide registry that has recruited 21 902 consecutive patients with acute cerebrovascular events from 132 hospitals in China between September 2007 and August 2008,[18] all of whom were assumed to receive intravenous tPA treatment, whereas all patients in the intervention arm received mechanical thrombectomy treatment. The total costs and quality-adjusted life-years (QALYs) that were gained with each alternative type of treatment were estimated for each health state at 90 days from the index event, at a 9-month period following this time point and then estimated annually for the remaining 30 years. The half-cycle correction was conducted for the years spent in the corresponding states that were subsequently used to calculate the health outcomes and costs. The transition probability and costs were discounted for the first 3-month segment and for the next 9-month segment in the first year. This analysis was conducted from the

perspective of healthcare payers. The present study used published data and anonymised clinical data of patients from databases and therefore was exempt from institutional review board approval.

### Input parameters

The model input parameters were drawn from the published literature and the claims database (table 1). The proportion of efficacy and safety outcomes at 90 days in the intravenous tPA group was estimated by meta-analysis based on a random effect model ($I^2 > 50\%$) of the recently published nine trials, including the Multicentre Randomized Clinical Trial of Endovascular Treatment for Acute Ischaemic Stroke in the Netherlands,[2] the Endovascular Treatment for Small Core and Anterior Circulation Proximal Occlusion with Emphasis on Minimizing CT to Recanalisation Times,[3] the Randomized Trial of Revascularization with Solitaire FR Device versus Best Medical Therapy in the Treatment of Acute Stroke Due to Anterior Circulation Large Vessel Occlusion Presenting within Eight Hours of Symptom Onset,[4] the Extending the Time for Thrombolysis in Emergency Neurological Deficits-Intra-Arterial,[5] the Solitaire with the Intention for Thrombectomy as Primary Endovascular Treatment (SWIFT PRIME),[6] the THRombectomie des Artères CErebrales,[19] The Randomized, Concurrent Controlled Trial to Assess the Penumbra System's Safety and Effectiveness in the Treatment of Acute Stroke,[20] the Pragmatic Ischaemic Thrombectomy Evaluation[21] and the Endovascular Acute Stroke Intervention.[22] The efficacy and safety outcomes ORs of mechanical thrombectomy-treated patients at day 90 were estimated by meta-analysis based on a fixed effect model ($I^2 < 50\%$) of the aforementioned nine trials (table 1). Subsequently, the proportions of the outcomes in the mechanical thrombectomy-treated groups were calculated based on the proportions in the intravenous tPA group and the ORs for the outcomes measured, as determined by the following formula: p2=(OR*p1)/(1+(OR–1)*p1).

The recurrent rates of stroke by modified Rankin Scale (mRS) categories and the death rate with recurrent strokes in years following the first 90 days were estimated from the CNSR study.[18] We further assumed an increase in the risk of stroke recurrence by 1.03-fold per life year.[23] The patients who remained alive after stroke recurrence were assumed to be reallocated equally among categories of equal and greater disability.[24] This indicated that patients in the independent state (mRS 0–2) who had a recurrent stroke and survived were allocated equally among the independent (mRS 0–2) and the dependent states (mRS 3–5), while patients in the dependent state remaining alive after stroke recurrence were all remain in dependent state.

The age-specific non-stroke death rates were derived from the most recent published census of China and were adjusted according to the causes of death of 2013 reported in the China Health Statistics Yearbook 2014.[25 26] The disability status was assumed to affect the survival rate, and therefore the final age-specific non-stroke death rates for those with dependent state were adjusted by the HR of death for mRS 3–5.[27]

### Costs

The total costs including both out-of-pocket costs and reimbursements were converted to 2013 Chinese yuan renminbi (CNY) using the medical care component of consumer price index.[26] The average cost of one-time hospitalisation and annual posthospitalisation costs (such as inpatient and outpatient rehabilitation and secondary preventive costs) after stroke according to the categories of mRS were obtained from the database of the CNSR study.[18] The additional costs of mechanical thrombectomy, including costs for devices, procedure and special periprocedural care, were estimated using the data from the EAST study and the Thrombolysis Implementation and Monitor of acute ischaemic Stroke in China (TIMS-CHINA; a national prospective registry of 1440 acute ischaemic stroke patients with thrombolytic therapy with intravenous tPA recruited from 67 centres in China between May 2007 and April 2012) study.[17 28] The additional costs of tPA treatment and occurrence of sICH were estimated using the data from the CNSR and the TIMS-CHINA studies. We did not include the indirect economic costs such as lost work productivity in the present study. All costs and utilities were discounted by 3% per year.[29]

### Health states

Patients could undergo transitions between the three health states according to the functional outcome based on mRS: independent (mRS 0–2), dependent (mRS 3–5) or deceased (mRS 6).[7 30] At the end of each Markov cycle, the patients either remained in their current health state, attained a health state with greater disability due to recurrent stroke or did not survive due to a recurrent stroke or a non-stroke cause (figure 1).

### Outcome assessment

Health outcomes were measured in terms of QALYs, which were calculated by multiplying the length of life by utility scores derived from the literature.[9 31 32] The utility scores of the different disability states after stroke were developed using the European quality of life scale European quality of life scale-5 dimensions (EQ-5D) along with the Chinese preference weights in a Chinese stroke population.[32] In the current model, the events of recurrent stroke and sICH were considered temporary health states unless they resulted in death. All patients who entered these health states were assumed to have a short-term disutility of 30 days for an event of recurrent stroke and only 14 days for an event of sICH.[9 31] The total economic costs were calculated by multiplying the number of patients in each state by the direct medical costs for that state. The incremental cost-effectiveness ratio (ICER) was calculated by dividing the difference of the costs by the difference in QALYs between the two

**Table 1** Base case and plausible ranges of model inputs

| Model input | Base case | Range | Reference |
|---|---|---|---|
| Efficacy and safety outcome inputs | | | 2–6 19–22 |
| Proportion of outcomes at 90 days in intravenous tPA group | | | |
| mRS 0–2 | 0.325 | 0.258–0.392 | |
| Death (mRS 6) | 0.168 | 0.131–0.205 | |
| sICH | 0.058 | 0.035–0.095 | |
| OR at 90 days | | | |
| mRS 0–2 | 2.046 | 1.692–2.474 | |
| Death (mRS 6) | 0.871 | 0.684–1.109 | |
| sICH | 0.965 | 0.665–1.399 | |
| Probabilities inputs | | | |
| Proportion of patients received mechanical thrombectomy | 0.861 | 0.839–0.883 | 2–6 19–22 |
| Proportion of patients arrived within 4.5 hours | 0.854 | 0.839–0.869 | CNSR |
| Recurrent rate of stroke (per patient year) | | | CNSR |
| mRS 0–2 | 0.1026 | 0.0961–0.1093 | |
| mRS 3–5 | 0.1418 | 0.1303–0.1534 | |
| Relative risk of stroke recurrence per life year | 1.03 | 1.02–1.04 | 23 |
| Death with recurrent stroke | 0.2101 | 0.1887–0.2316 | CNSR |
| Age specific non-stroke death rate* | 0.0089–0.1653 | | 25 26 |
| HR of non-stroke death for mRS 3–5 | 1.78 | 1.43–2.14 | 27 |
| Cost inputs (2013 Chinese yuan renminbi) | | | |
| Additional costs of mechanical thrombectomy | 60821 | 52314–70311 | EAST |
| Additional costs of intravenous tPA treatment | 11179 | 10555–11829 | CNSR, TIMS-CHINA |
| Additional costs of sICH | 2374 | 2249–2504 | TIMS-CHINA |
| One-time hospitalisation costs | | | CNSR |
| mRS 0–2 | 10055 | 9907–10205 | |
| mRS 3–5 | 13729 | 13428–14035 | |
| mRS 6 | 11121 | 10219–12081 | |
| Annual posthospitalisation costs | | | CNSR |
| mRS 0–2 | 7385 | 7156–7619 | |
| mRS 3–5 | 11350 | 10730–11996 | |
| Utility inputs | | | |
| mRS 0–2 | 0.76 | 0.69–0.82 | 32 |
| mRS 3–5 | 0.21 | 0.17–0.26 | 32 |
| Death (mRS 6) | 0 | 0.00–0.00 | 32 |
| Recurrent stroke | 0.34 | 0.32–0.36 | 9 |
| sICH | 0.84 | 0.72–1.0 | 31 |
| Discount rate inputs | | | |
| Costs | 0.03 | 0.03–0.08 | 29 |
| Outcomes | 0.03 | ±20% | 29 |

All costs were converted to 2013 Chinese yuan renminbi by using the medical care component of consumer price index; to convert CNY to US$, divide by 6.5.

*Age-specific non-stroke death rate: only the number of 63 years old (0.0089) and 93 years old (0.1653) are presented.

CNSR, China National Stroke Registry; EAST, Endovascular therapy for Acute ischaemic Stroke Trial; mRS, modified Rankin Scale; sICH, symptomatic intracerebral haemorrhage; TIMS-CHINA, Thrombolysis Implementation and Monitor of acute ischaemic Stroke in China; tPA, tissue plasminogen activator.

treatment alternatives. We modelled the costs and QALYs gained over the short-term (1, 5 and 6 years) and the long-term (30 years). The mechanical thrombectomy was considered cost-effective if the ICER was less than CNY 125 700 (3× gross domestic product (GDP) per capita of China in 2013,[26] US$ 19 300) per QALY gained. This willingness-to-pay threshold was recommended by the Commission on Macroeconomics and Health of WHO.[29]

## Sensitivity analysis

A deterministic one-way sensitivity analysis that used varying probabilities, utilities and costs was conducted in order to evaluate the uncertainty of the long-term (30 years) results of the model. The variation in these parameters was conducted once at a time at the plausible rages (table 1). In order to determine how much worse mechanical thrombectomy could have performed but still produced a cost-effective ICER, we constructed the two hypothetical worse outcomes for the performance of the mechanical thrombectomy. This was accomplished by setting the ORs of mRS 0–2 at day 90 for the mechanical thrombectomy treatment at the lower limits of 90% and 95% CI in the meta-analyses of the nine trials. This represented the unfavourable and worse unfavourable scenarios, respectively. For each outcome scenario, we further constructed four hypothetical cost scenarios by setting 10% increase or 10%, 25% and 50% decrease of the costs of mechanical thrombectomy.

Furthermore, a probabilistic sensitivity analysis was further undertaken in order to evaluate the stochastic uncertainty due to the simultaneous variability of the variables examined. It was conducted by using Monte Carlo simulation in Ersatz V.1.3 (a bootstrap add-in for Microsoft Excel for Windows; EpiGear International, Brisbane, Australia). We assumed that costs followed a log-normal distribution and that the probabilities and utilities followed a beta distribution. The simulation was run 10 000 times. The results were summarised using a scatter plot and a cost-effectiveness acceptability curve.

## RESULTS

### Base case analysis

Table 2 indicates the costs, outcomes and ICER for the mechanical thrombectomy treatment calculated in the short term (1, 5 and 6 years) and in the long term (30 years). In the base case scenario, for a 63-year-old patient with acute ischaemic stroke caused by proximal anterior circulation occlusion within 6 hours after onset of stroke, mechanical thrombectomy would be cost-ineffective in the first 5 years, but become cost-effective from the sixth year onwards, using the threshold of CNY 125 700 (3× GDP per capita of China in 2013, US$19 300) as the willingness-to-pay per QALY. After 6 years, the mechanical thrombectomy gained 0.430 QALYs at an additional cost of CNY 48 940 (US$7530), yielding an ICER of CNY 113 800 (US$17 510) per QALY gained. In the long term (30 years), mechanical thrombectomy gained 0.794 QALYs at an additional cost of CNY 50 000 (US$7700), yielding an ICER of CNY 63 010 (US$9690) per QALY gained.

### Sensitivity analysis

The results of the deterministic one-way sensitivity analysis for the ICER of the mechanical thrombectomy in the long term are presented in the tornado diagrams (figure 2). Overall, the ICER was most sensitive to the OR of the favourable functional outcome (mRS 0–2) at day 90, additional cost of mechanical thrombectomy and utility of mRS 0–2. In case of a decrease in the OR of mRS 0–2 at day 90 to 1.692, the ICER of the mechanical thrombectomy (CNY 87 123/QALY) was still within the threshold of the willingness-to-pay per QALY (CNY 125 700, 3× GDP per capita of China in 2013). In each hypothetical case scenario, the mechanical thrombectomy continued to produce a benefit in QALYs (online supplementary table 1). Even in the worst hypothetical scenario with worse unfavourable effect outcome and an increase of 10% price of the mechanical thrombectomy, this type

| Table 2 | Costs and outcomes in base case analysis | | | |
|---|---|---|---|---|
| **Time horizon** | **Treat strategy** | **QALYs** | **Cost (CNY)** | **ICER (CNY/QALY)** |
| 1 year | Intravenous tPA alone | 0.326 | 27 220 | – |
| | Mechanical thrombectomy+intravenous tPA | 0.405 | 77 700 | 638 987 |
| 5 years | Intravenous tPA alone | 1.392 | 58 590 | – |
| | Mechanical thrombectomy+intravenous tPA | 1.765 | 107 710 | 131 689 |
| 6 years | Intravenous tPA alone | 1.599 | 65 230 | – |
| | Mechanical thrombectomy+intravenous tPA | 2.029 | 114 170 | 113 814 |
| 30 years | Intravenous tPA alone | 2.979 | 117 940 | – |
| | Mechanical thrombectomy+intravenous tPA | 3.773 | 167 970 | 63 010 |

ICER, incremental cost-effectiveness ratio; QALY, quality-adjusted life-year; tPA, tissue plasminogen activator.

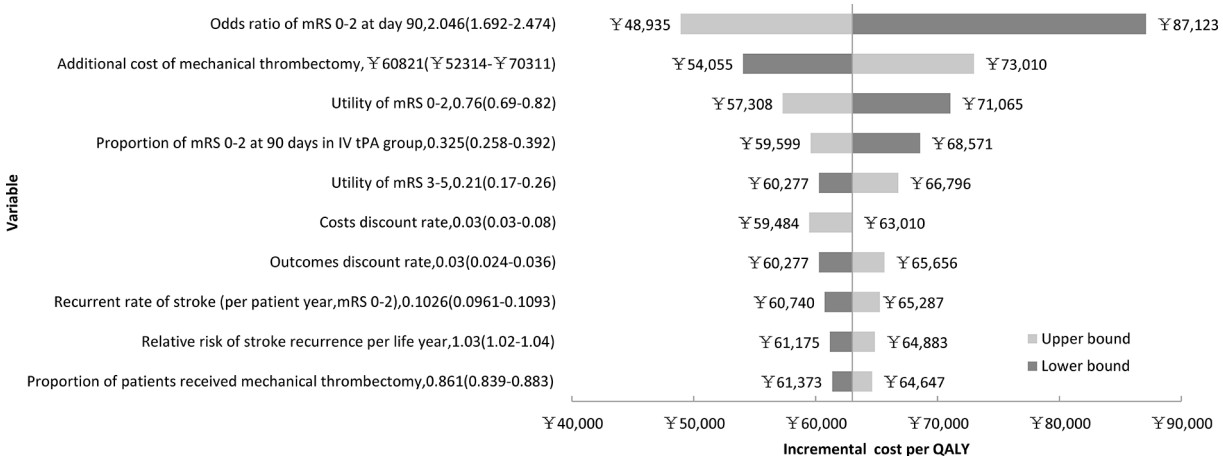

**Figure 2** One-way sensitivity analysis on incremental cost-effectiveness ratio (ICER) gained in the long term (30 years) by mechanical thrombectomy. All model input parameters were analysed, and only 10 parameters with the highest relative effects on ICER are displayed. Base case scenario of ICER is CNY 63 010 per quality-adjusted life-year (QALY) gained. CNY, Chinese yuan renminbi; IV, intravenous; mRS, modified Rankin Scale; tPA, tissue plasminogen activator.

of treatment continued to be cost-effective (ICER: CNY 95 839/QALY). In the hypothetical case scenario with a base case effect outcome and a decrease of 50% in the price of mechanical thrombectomy, this type of treatment could be highly cost-effective (ICER: CNY 30 995/QALY <CNY 41 900 [1× GDP per capita of China in 2013, US$6400]/QALY).

Figure 3 indicates the results of the probabilistic sensitivity analysis in the long term with parameters of the model inputs presented in the online supplementary table 2. Among the 10 000 simulation runs, mechanical thrombectomy was cost-effective in 99.9% of the simulations at a willingness-to-pay threshold of CNY 125 700 (3× GDP per capita of China in 2013, US$19 300) per QALY. The online supplementary figure 1 indicates the

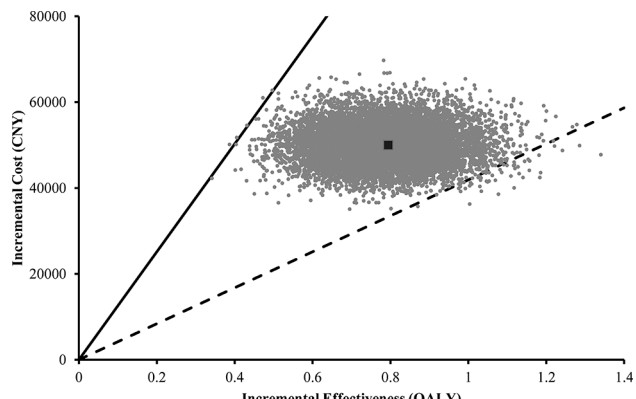

**Figure 3** Scatter plot of the result of probabilistic sensitivity analysis in the long term (30 years). Each point represents a simulation. The dark square represents the base case (0.794 QALYs gained at an incremental cost of CNY 50 000). The solid line represents the willingness-to-pay threshold of CNY 125 700 per QALY. The dashed line represents CNY 41 900 per QALY. Points to the right of the solid line are considered cost-effective. CNY, Chinese yuan renminbi; QALYs, quality-adjusted life-years.

cost-effectiveness acceptability curve of the mechanical thrombectomy.

## DISCUSSION

The present study indicated that mechanical thrombectomy with second-generation devices for acute ischaemic stroke that was caused by proximal anterior circulation occlusion was cost-effective from the sixth year onwards in China. Each patient with acute ischaemic stroke treated with mechanical thrombectomy gained an ICER of CNY 63 010 per QALY in the long term, which was below 3× GDP per capita of China in 2013 (CNY 125 700). The study further demonstrated that the ICER was more sensitive to the OR of favourable functional outcome at day 90, additional cost of mechanical thrombectomy and the utility for patients with independence.

The present study indicated that mechanical thrombectomy treatment was cost-effective in low-income and middle-income countries such as China, which is a similar conclusion to that completed in other high-income countries.[7–10] The lifetime gain of 0.794 QALYs with regard to the mechanical thrombectomy treatment for acute ischaemic stroke in this study was comparable with that reported in the high-income countries (0.7 QALYs in the USA,[7] 1.05 QALYs in the UK[9] and 0.99 QALYs in Sweden[10]). The QALYs gain of mechanical thrombectomy treatment was relatively higher compared with that of the majority of the other treatments for stroke. For example, the lifetime QALYs gain was 0.42 for intravenous tPA treatment for acute ischaemic stroke within 4.5 hours,[33] and 0.17 for clopidogrel for secondary prevention of stroke compared with aspirin.[34] This may be due to the large magnitude of the effect of the mechanical thrombectomy (OR 2.046).

Mechanical thrombectomy treatment with second-generation devices has been accepted as the standard of care for patients with acute ischaemic stroke caused by proximal anterior circulation occlusions within 6 hours

after symptom onset.[35 36] However, economic costs of the mechanical thrombectomy treatment are extremely high, especially in low-income and middle-income countries such as China, where people experience a higher incidence of stroke and higher prevalence of intracranial atherosclerosis compared with the Western countries.[37] The additional cost of the mechanical thrombectomy was approximately fivefold higher compared with that noted for the one-time hospitalisation without tPA treatment and mechanical thrombectomy in China. This difference between the two aforementioned parameters was only 1.5-fold in the USA.[7 8] Previous studies indicated socioeconomic disparities with regard to the utilisation of mechanical thrombectomy for acute ischaemic stroke and the patients with low income who were resident in remote areas exhibited lower rates of mechanical thrombectomy utilisation.[38 39] The implementation of mechanical thrombectomy treatment was, to some extent, dependent on the cost-effectiveness of the technology, which is particularly significant for the clinical decision in the low-income and middle-income countries. The present study supported the implementation of the mechanical thrombectomy treatment after acute ischaemic stroke in clinical practice in the low-income and middle-income countries from the perspective of economics. The data may provide an important reference for the low income and/or remote areas in the Western countries.

Proximal large vessel atherosclerotic stenosis or occlusion accounts for 35%–40% of all acute ischaemic strokes,[17] among which approximately 40% are admitted to the hospital within 6 hours in China.[18] Therefore, approximately 14%–16% of patients with ischaemic stroke were eligible for mechanical thrombectomy and might benefit from this procedure. However, the real-world implementation of endovascular thrombectomy treatment in low-income and middle-income countries and areas may be restricted by the poor awareness of the public, poor infrastructure, inefficient systems, deficiency of specialists and the time points of patient entry to the hospital (within 6 hours), which may cause inequity for those who cannot receive the technology.[10] Thus, the education that is targeted to the public, hospital administration and governmental agencies should be improved so that users can fully understand the benefit and cost-effectiveness of thrombectomy.[40] Furthermore, the service system redesign is required to establish efficient care chains and workflow with coordination between neurointerventinalists and other departments. Additionally, high experience and skills are required to perform this advanced technology, while the ways of providing interventional treatment are non-unstandardised and diverse in low-income and middle-income countries such as China.[40] Future studies should focus on the organisation of standardised training for performing thrombectomy in order to compensate for the deficiency of intervention specialists in these countries and areas.

The current study has several limitations that should be considered when interpreting the results. First, our model focused on the impact of mechanical thrombectomy treatment on acute ischaemic stroke, and the health status and costs that were involved as a result of other causes, such as occurrence of intracranial haemorrhage and myocardial infarction, were not included in this model. Second, the costs of transfer to the hospitals doing mechanical thrombectomy were not included in this analysis. Third, functional improvement after rehabilitation was not considered in the model due to the lack of available data on the efficacy of rehabilitation. However, organised rehabilitation after stroke in China is poor.[41 42] Fourth, the current study arbitrarily assumed to some extent that the patients with independent state remaining alive after stroke recurrence were equally reallocated in the categories of independent and dependent states. However, this is not unprecedented in the modelling of cost-effectiveness analysis for stroke.[24] Finally, the efficacy of mechanical thrombectomy treatment was based on trials that were completed in high-income countries with older age of participants compared with those reported in China. In addition, the majority of the trials were terminated early and all were sponsored by the industry, which may have caused potential risk of bias, whereas some of the patients included in the trials did not receive intravenous tPA treatment. However, the current analysis included all data that were possibly available. These limitations would have led to underestimation or overestimation of the true cost-effectiveness of mechanical thrombectomy treatment in low-income and middle-income countries.

## CONCLUSIONS

Mechanical thrombectomy with second-generation devices for acute ischaemic stroke caused by proximal anterior circulation occlusion within 6 hours after the onset of stroke was cost-effective in China. The current study supports the implementation of mechanical thrombectomy treatment after acute ischaemic stroke in clinical practice in low-income and middle-income countries and may also be a reference to the low income and/or remote areas in the developed countries. Additional medical resources that are related to mechanical thrombectomy should be allocated in these areas.

**Contributors** YP, YoW and YiW designed the study and drafted the manuscript. YP, XC and XH: collected the data, performed the literature search and constructed the decision tree. XZ, LL and ZM interpreted the data and revised the manuscript.

**Funding** This work was supported by grants from the Ministry of Science and Technology of the People's Republic of China (2015BAI12B04, 201 5BAI12B02, 2016YFC0901000, 2016YFC0901001 and 2016YFC0901002), grants from Beijing Municipal Science and Technology Commission (Z151102200390000 and Z151100003915117), and grants from Beijing Municipal Commission of Health and Family Planning (No. 2016-1-2041, SML20150502).

**Competing interests** None declared.

**Patient consent** Obtained.

**Provenance and peer review** Not commissioned; externally peer reviewed.

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
