## [Reviewer comments · BMJ Open]

ARTICLE DETAILS

TITLE (PROVISIONAL)	Cost-Effectiveness of Mechanical Thrombectomy within 6 Hours of Acute Ischaemic Stroke in China
AUTHORS	PAN, YUESONG; Cai, Xueli; Huo, Xiaochuan; Zhao, Xingquan; Liu, Liping; Wang, Yongjun; Miao, Zhongrong; Wang, Yilong

VERSION 1 – REVIEW

REVIEWER	Dr Joyce Balai Centre for Evidence-Based Medicine. University of Oxford United Kingdom DPhil Candidate in Evidence-Based Health Care and working on micro-costing of mechanical thrombectomy for acute ischaemic stroke.
REVIEW RETURNED	05-Sep-2017

GENERAL COMMENTS	The authors present an economic evaluation of the cost-effectiveness of mechanical thrombectomy based on the evidence from the recent RCTs on MT trials. Despite the limitations, they have a long interval follow-up and adhered to the CHEERS checklist. This is an important paper for clinicians, health economics, policy makers and healthcare providers, particularly in developing countries. However, there are some comments that need addressing: Major Comments: 1) There is a need for a clear statement of the study aim and research question in the background.2) A population statistics would be useful and where possible an estimate of the number of people that might benefit from this procedure (i.e. the percentage of all strokes that could be eligible mechanical thrombectomy. This would give the reader a target size for the usefulness of this new intervention.3) Can the authors comment on, if any the implications of their study for practice, policy and future research in the conclusion?4) The individual patient data analysis includes MT with or without IVT vs IVT and/or best medical therapy (only SWIFT PRIME and EXTEND-IA mandated IV tPA in the inclusion criteria, while ESCAPE, MR CLEAN and REVASCAT tested MT against “best medical management” that may or may not include IV tPA) and their decision tree includes MT+IVT vs IVT alone. A) The authors might comment on whether this is a valid review to make a comparison. B) is it possible for the authors to calculate the cost-effectiveness for MT with best medical treatment without IVT?
---

	5) Since this cost-effectiveness of mechanical thrombectomy applies to developing countries, then why not use the World Health Organization Quality of Life (WHOQOL) instrument, which cuts across all cultures and more applicable to developing countries than the European quality of life scale (EQ-5D), which is more for developed countries. Minor comments: 1) The authors should be consistent with spellings i.e. British or American, but suggest British spelling, for example, should be Ischaemia, recanalisation, haemorrhage etc. Similarly, there are a couple of typos 2) The authors could consider updating their references with the following papers on the cost-effectiveness of stent retrievers: a) Fernando de Andre´s-Nogales and colleagues. Cost-effectiveness of mechanical thrombectomy using stent retriever after intravenous tissue plasminogen activator compared with intravenous tissue plasminogen activator alone in the treatment of acute ischaemic stroke due to large vessel occlusion in Spain. European Stroke Journal 2017, Vol. 2(3) 272–284 b) Shireman et al. 2017: Cost-Effectiveness of Solitaire Stent Retriever Thrombectomy for Acute Ischemic Stroke Results From the SWIFT-PRIME Trial (Solitaire with the Intention for Thrombectomy as Primary Endovascular Treatment for Acute Ischemic Stroke. Stroke. 2017; 48:379-387.
--	--

REVIEWER	Neeraj Chaudhary Michigan Medicine USA
REVIEW RETURNED	09-Sep-2017

GENERAL COMMENTS	Mathematical analyses are needed for budget justification for providing new age care in acute ischemic stroke. However, mathematical models cannot replace clinical data. Wherever, hospitals in China can provide the stroke care in question, should pool data together with follow up analysis to answer the wider question of applicability of new age care with higher up front cost but with an ultimate cost saving via improvement in patient outcome and consequent decreased burden of cost to society. The submitted manuscript is too presumptive and assumptive with mathematical analysis devoid of any clinical data points. Hence the manuscript in my opinion does not merit acceptance to publish
---

REVIEWER	Conor Teljeur Trinity College, University of Dublin Ireland
REVIEW RETURNED	14-Nov-2017

GENERAL COMMENTS	Overall the paper is well laid out and reasonably well written. In places the text could do with grammatical improvement. I have some substantial concerns about the manner in which parameters have been defined for the economic model, although it is possible that these can be relatively simply addressed. Specific comments:
---

Background

The introduction makes no reference to first and second generation devices and the difference in efficacy measured in the trials. It would be useful for context as the data from those first three trials is subsequently excluded in the analysis - some explanation would help.

I am surprised the Canadian health technology assessment (Xie X, et al. Mechanical thrombectomy in patients with acute ischemic stroke: a cost-utility analysis. CMAJ Open. 2016;4(2):E316-E25) has not been referenced.

Methods - model overview:

The mean age of Chinese patients is quoted as 63 years, which is somewhat less than the mean age of trial participants (68). Is there a reason for that and should we be concerned over the applicability of the trial data to the Chinese setting?

It is stated that all patients in the intervention arm received mechanical thrombectomy. This was not the case in the trials, where less than 90% on average received the intervention [as recanalisation had been successfully achieved through IV-tPA alone]. By restricting to only those who got mechanical thrombectomy, the reported trial outcomes are no longer applicable.

To counter the above point, patients who received mechanical thrombectomy sometimes had more than one device used in the procedure. I think the data used by Garasalingam reports 1.2 devices per patient. These issues are important from a costing point of view.

Methods - input parameters:

Why use the individual patient data meta-analysis of five trials when there are data from nine available? Meta-analyses including the later trials have found a slightly lower effect, and excluding them for no good reason could be interpreted as introducing unnecessary bias. I cannot see what you have used from the IPD meta-analysis that justifies selecting over an above the full set of trials. Unless you can provide strong justification it should be revised.

I would like to see some more detail about the transition probabilities for the period after 90 days post-stroke. The model uses a 30 year time horizon - so patients are modeled to age 93. This raises two issues: what is life expectancy for a healthy 63 year old Chinese person and what is the likely life expectancy for a stroke survivor; and do you have 30 year follow up data on stroke survivors that makes you comfortable that some might live that long? Other economic analyses have used much shorter time horizons on the grounds that there was insufficient evidence about longer term outcomes. Remember, you are trying to model 30 year outcomes based on 90 day outcome data - that is a bit of a stretch and I would prefer if the main analysis went to no more than 10 years.

Methods - costs:

It is unclear what costs are being included in the annual post-hospitalisation costs. For those with mRS 3-5, they are functionally dependent and likely to have substantial care needs. In the model it includes an annual cost equivalent to US€1,746 for this cohort. Is this the cost the accrues to the payer, and the patient and their family then pay out-of-pocket for the rest?

It seems an incredibly low cost to care for a functionally dependent patient for 12 months. If a very substantial cost falls on the patient and family then that must be clearly highlighted, and there should be justification for not including a societal perspective. In Ireland we were looking at annual costs of €26,604 (more than US\$30,000) for care of mRS3-5 patients.

Methods - health states:
How long were the Markov cycles?

Were transition probabilities static by Markov cycle?

What were the transition probabilities?

Methods - outcome assessment:
How did you decide on the uncertainty for the utility parameters?

The figure used for mRS3-5 seems very low - looking at the various data sources used in other economic evaluations of mechanical thrombectomy suggests a figure between 0.3 and 0.4, rather than 0.21. I'd like to see better justification for the figures used.

Methods - sensitivity analysis:
Is the main analysis based on a deterministic model or are the summary figures derived from the probabilistic analysis?

It states in the text that the costs followed log normal distributions, and yet the ranges shown in Table 2 are left- rather than right-skewed. It does not make any sense. Please explain the distribution parameters and why the skew is in the wrong direction.

Results:
Due to some of the issues in how the input parameters have been defined and queries over the applicability of the cost data, I cannot comment on the results in detail, although I am concerned at the use of a 30 year time horizon. Ideally the tornado plot would be for the 6 year time horizon.

Discussion:
One issue is that patients need to get to the hospital within that 6 hour time frame for mechanical thrombectomy. Is the analysis restricted to those who can get to the hospital in that time? It would have to be - in which case you should discuss the inequity for patients who cannot get there in time - they will be discriminated against. Furthermore - will mechanical thrombectomy be provided at all hospitals that provide IV-tPA?

If not, then there will be a centralisation of services and some patients will require transfer to the centre doing mechanical thrombectomy - those costs have not been included.

All or almost all of the second generation trials ceased early and were sponsored by industry - are these points not worthy of a mention? They point to potential risk of bias that should be highlighted.

The limitations need to address the use of a limited set of the available trials and clearly explain why the other four trials were not used in evidence synthesis.

	I think you should remove reference to the results being robust as shown in sensitivity analyses. The appearance of robustness is a function of structural uncertainty and how you have defined parameter uncertainty. I am unconvinced that either has been adequately addressed. Table 1: Was there no uncertainty around the proportion patients achieving mRS0-2, mRS3-5, mRS6, and sICH? There is substantial uncertainty around the risk ratios for these outcomes, and therefore there should be uncertainty in the model about the efficacy of mechanical thrombectomy. Table 2: Why is there no uncertainty around the proportion of patients arriving within 4.5 hours? The ranges for the uncertainty on utility inputs for mRS0-2, mRS3-5, and sICH are specified incorrectly - they all skew in the wrong direction. A beta distribution with a mean greater than 0.5 will be either symmetrical or left skewed (and a beta with mean <0.5 will be symmetrical or right skewed). So the mRS0-2 range should be something like 0.60 to 0.89, not 0.60 to 0.96. How were these parameters specified? Given that equal discounting is used for costs and benefits in the main analysis, why was equal discounting not applied in the sensitivity analysis (ie same lower and upper bound applied to both simultaneously)? Please justify.
--	---

VERSION 1 – AUTHOR RESPONSE

Response to Reviewers' Comments

Reviewer #1:

The authors present an economic evaluation of the cost-effectiveness of mechanical thrombectomy based on the evidence from the recent RCTs on MT trials. Despite the limitations, they have a long interval follow-up and adhered to the CHEERS checklist. This is an important paper for clinicians, health economics, policy makers and healthcare providers, particularly in developing countries. However, there are some comments that need addressing:

Major Comments:

3. There is a need for a clear statement of the study aim and research question in the background.

Reply: We thank for the reviewer's comments. We have revised the last sentence in the Background section (at the bottom of Page 4) and clearly stated the study aim.

4. A population statistics would be useful and where possible an estimate of the number of people that might benefit from this procedure (i.e. the percentage of all strokes that could be eligible mechanical thrombectomy. This would give the reader a target size for the usefulness of this new intervention.

Reply: Thanks for the reviewer's comments. According to previous studies, proximal large vessel atherosclerotic stenosis or occlusion accounts for 35–40% of all acute ischaemic strokes (Stroke Vasc Neurol. 2016;1:44-51), among which about 40% arrived at the hospital within 6 hours (data from the CNSR study).

About 14%-16% of ischaemic stroke patients were eligible mechanical thrombectomy and might benefit from this procedure. We have added this issue in the Discussion section in the revised manuscript (2nd paragraph in Page 12).

5. Can the authors comment on, if any the implications of their study for practice, policy and future research in the conclusion?

Reply: We thank for the reviewer's comments. The implication of this study for clinical practice and policy was stated in 3rd and 4th paragraphs of the Discussion section. To emphasize the implication of the study, we have revised the last sentence in the conclusion to "Our study supports the implement of mechanical thrombectomy treatment after acute ischaemic stroke in clinical practice in low- and middle-income countries, and may also be a reference to the low income or remote areas in the developed countries. More medical resources related to mechanical thrombectomy should be allocated in these areas." in the revised manuscript (Last sentence in the conclusion, Page 14).

6. The individual patient data analysis includes MT with or without IVT vs IVT and/or best medical therapy (only SWIFT PRIME and EXTEND-IA mandated IV tPA in the inclusion criteria, while ESCAPE, MR CLEAN and REVASCAT tested MT against "best medical management" that may or may not include IV tPA) and their decision tree includes MT+IVT vs IVT alone. A) The authors might comment on whether this is a valid review to make a comparison. B) is it possible for the authors to calculate the cost-effectiveness for MT with best medical treatment without IVT?

Reply: We thank for the reviewer's comments. We agree with the reviewer that the results of the analysis may influenced by the fact that some of the trials tested MT against "best medical management" that may or may not include IV tPA. However, most of the patients included in the trials received IV tPA. In this analysis we focused on MT+ IV tPA vs IV tPA alone since the guideline for stroke management recommended patients eligible for IV tPA should receive IV tPA even if endovascular treatments are being considered. Although it might be possible to calculate the cost-effectiveness for MT with best medical treatment without IV tPA, we would like to do the evaluation in future research. We added this issue in the Discussion section in the revised manuscript (1st paragraph in Page 14).

7. Since this cost-effectiveness of mechanical thrombectomy applies to developing countries, then why not use the World Health Organization Quality of Life (WHOQOL) instrument, which cuts across all cultures and more applicable to developing countries than the European quality of life scale (EQ-5D), which is more for developed countries.

Reply: We agree with the reviewer that WHOQOL is a useful instrument to estimate the quality of life for stroke patients in the developing countries. EQ-5D is also a common used instrument to measure quality of life of patients in stroke trials (Neurology. 2015;85(7):573-9.). And in our previous study (CNS Neurosci Ther 2014;20:1029-35), utility scores of different disability states were developed using EQ-5D along with the Chinese preference weights in a Chinese stroke population. Therefore, the utility developed using ED-5D was applicable to China, a developing country.

Minor comments:

8. The authors should be consistent with spellings i.e. British or American, but suggest British spelling, for example, should be- Ischaemia, recanalisation, haemorrhage etc. Similarly, there are a couple of typos

Reply: Thanks very much. We have revised throughout the manuscript using British spelling and corrected the typos.

9. The authors could consider updating their references with the following papers on the cost-effectiveness of stent retrievers:

a) Fernando de Andre´s-Nogales and colleagues. Cost-effectiveness of mechanical thrombectomy using stent retriever after intravenous tissue plasminogen activator compared with intravenous tissue plasminogen activator alone in the treatment of acute ischaemic stroke due to large vessel occlusion in Spain. *European Stroke Journal* 2017, Vol. 2(3) 272–284

b) Shireman et al. 2017: Cost-Effectiveness of Solitaire Stent Retriever Thrombectomy for Acute Ischemic Stroke Results From the SWIFT-PRIME Trial (Solitaire with the Intention for Thrombectomy as Primary Endovascular Treatment for Acute Ischemic Stroke. *Stroke*. 2017; 48:379-387.

Reply: Thanks very much. We have added these two references in the background of the revised manuscript (2nd paragraph in the Background, Page 4).

Reviewer #2 :

10. Mathematical analyses are needed for budget justification for providing new age care in acute ischemic stroke. However, mathematical models cannot replace clinical data. Wherever, hospitals in China can provide the stroke care in question, should pool data together with follow up analysis to answer the wider question of applicability of new age care with higher up front cost but with an ultimate cost saving via improvement in patient outcome and consequent decreased burden of cost to society.

The submitted manuscript is too presumptive and assumptive with mathematical analysis devoid of any clinical data points. Hence the manuscript in my opinion does not merit acceptance to publish.

Reply: We thank for the reviewer's comments. We agree with the reviewer that mathematical models cannot replace clinical data for budget justification for providing new age care in acute ischemic stroke. Clinical data, especially clinical trial provided the evidence of efficacy and safety of the technology. However, as we clarified in the Discussion section, the implement of new age care in clinical practice is to-some-extent depend on the cost-effectiveness of the technology, which is of importance for the clinical decision in the low- and middle-income countries. We agree with reviewer that pool data together with long-term follow up analysis may answer the wider question of applicability of new age care. However, long-term follow-up data are always scarce for patients with acute stroke and with endovascular treatment. Acutely, cost-effectiveness analysis with mathematical models were common conducted for new technology of stroke care after clinical trials (*Stroke*. 2013; 44(8):2269-74. *Neurology*. 2016; 86(11):1053-9. *Stroke*. 2016; 47(11):2797-2804.). Like other previous studies on cost-effectiveness analysis of stroke care using mathematical models, this study evaluated the cost-effectiveness of mechanical thrombectomy for acute stroke and supported the implement of mechanical thrombectomy treatment after acute ischemic stroke in clinical practice in low- and middle-income countries. This could be of importance for clinical applicability of the technology in low- and middle-income countries from the perspective of economics.

Reviewer #3:

Overall the paper is well laid out and reasonably well written. In places the text could do with grammatical improvement. I have some substantial concerns about the manner in which parameters have been defined for the economic model, although it is possible that these can be relatively simply addressed.

Specific comments:

11. Background:

The introduction makes no reference to first and second generation devices and the difference in efficacy measured in the trials. It would be useful for context as the data from those first three trials is subsequently excluded in the analysis - some explanation would help.

Reply: We thank for the reviewer's comments. In this study, we referred to the second-generation devices and excluded the first three trials with first-generation devices. We clarified this issue in the introduction in the revised manuscript (Page 4).

12. I am surprised the Canadian health technology assessment (Xie X, et al. Mechanical thrombectomy in patients with acute ischemic stroke: a cost-utility analysis. *CMAJ Open*. 2016;4(2):E316-E25) has not been referenced.

Reply: Thanks very much. We have added the reference in the background of the revised manuscript (2nd paragraph in the Background, Page 4).

13. Methods - model overview:

The mean age of Chinese patients is quoted as 63 years, which is somewhat less than the mean age of trial participants (68). Is there a reason for that and should we be concerned over the applicability of the trial data to the Chinese setting?

Reply: The reviewer is right. The average age of hospitalized stroke patients in China is about 63 years, which is about 5 years lower than that of the Western countries. The average age in this study (63 years) is similar with other nationwide multicenter studies performed in China (63.8 years in the CNSR study: *Int J Stroke*. 2011;6(4):355-61.) (63 years in China-QUEST study: *Stroke*. 2009;40(6):2149-56.) (62 years in the CHANCE study: *N Engl J Med*. 2013;369(1):11-9.). The reason of this disparity is still unclear.

We agree with the reviewer that it may be unsuitable to use the efficacy of mechanical thrombectomy treatment based on trials with older age of patients. However, no large-scale clinical trial on the efficacy and safety of mechanical thrombectomy was performed in China and this is the only data available. These limitations would have led to under- or over- estimation of the true cost-effectiveness of mechanical thrombectomy treatment in China. We clarified this issue as a limitation of this study (2nd paragraph in Page 13).

14. It is stated that all patients in the intervention arm received mechanical thrombectomy. This was not the case in the trials, where less than 90% on average received the intervention [as recanalisation had been successfully achieved through IV-tPA alone]. By restricting to only those who got mechanical thrombectomy, the reported trial outcomes are no longer applicable.

To counter the above point, patients who received mechanical thrombectomy sometimes had more than one device used in the procedure. I think the data used by Garasalingam reports 1.2 devices per patient. These issues are important from a costing point of view.

Reply: You are right. We agree with the reviewer that not all the patients received the intervention in the trials. We added a parameter of proportion of patients received mechanical thrombectomy with the 95% confidence intervals as the uncertainty in the model and updated the data throughout in the revised manuscript (see table 1). After adding this input, the final ICER slightly reduced.

We also agree with the reviewer that some patients may receive more than 1 devices in the procedure and this may influence the costs. However, our data of costs were calculated through total average treatment costs recorded in a multicenter study which considered the issue of numbers of devices used in the procedures.

15. Methods - input parameters:

Why use the individual patient data meta-analysis of five trials when there are data from nine available? Meta-analyses including the later trials have found a slightly lower effect, and excluding them for no good reason could be interpreted as introducing unnecessary bias. I cannot see what you have used from the IPD meta-analysis that justifies selecting over an above the full set of trials. Unless you can provide strong justification it should be revised.

Reply: We thank for the reviewer's comments. It is an important issue. The meta-analyses included 9 trials (J Am Coll Cardiol. 2015;66(22):2498-505) included trials using the first-generation thrombectomy devices, such as the MR RESCUE trial (2013) and the IMS III trial (2013). Trials that completed in 2013, used first-generation thrombectomy devices and failed to demonstrate clinical benefit compared with IV-tPA. However, second-generation retrievable stents achieve higher recanalization rates compared with first-generation thrombectomy devices. Trials that completed in 2015, used second-generation thrombectomy devices and demonstrate clinical benefit compared with IV-tPA. Because of the negative results of studies in 2013 and positive results of studies in 2015, we mostly used the second-generation thrombectomy devices after 2015 in clinical practices. Therefore, we consider meta-analyses included the 5 trials in 2015 could be more closed to the clinical practice in the future. We clarified this issue and emphasized that our study focused on the second-generation thrombectomy devices in the Background section in the revised manuscript (Page 4).

16. I would like to see some more detail about the transition probabilities for the period after 90 days post-stroke. The model uses a 30 year time horizon - so patients are modeled to age 93. This raises two issues: what is life expectancy for a healthy 63 year old Chinese person and what is the likely life expectancy for a stroke survivor; and do you have 30 year follow up data on stroke survivors that makes you comfortable that some might live that long? Other economic analyses have used much shorter time horizons on the grounds that there was insufficient evidence about longer term outcomes. Remember, you are trying to model 30 year outcomes based on 90 day outcome data - that is a bit of a stretch and I would prefer if the main analysis went to no more than 10 years.

Reply: We thank for the reviewer's comments. We did not have 30-year follow-up data on stroke survivors and the life-expectancy for a stroke survivor is also unavailable. We performed a lifetime (30 years) simulation of the model did not mean that all the patients in our model survived 30 years, but this time horizon covered most death within 30 years. Actually, according to simulations in our model, about half of patients survived 11 years and only about 1% survived 30 years.

We agree with the reviewer that it might be a bit of a stretch to try to model 30-year outcomes based on 90-day outcome data. However, long-term or life-time period was commonly used to simulate the cost-effectiveness of stroke care. Most previous published studies on cost-effectiveness analysis of mechanical thrombectomy for stroke used the life-time horizons in the model (30 years: Stroke. 2016;47(11):2797-2804. Stroke. 2015;46(7):1870-6; 25 years: Int J Stroke. 2017;12(8):802-814. Neurology. 2016;86(11):1053-9.). Therefore, we keep the results of 30-year time horizons to compare with the studies performed in other countries. Besides the 30-year time horizons, we also simulate the short-term results (4 years and 5 years) (Table 2 in the revised manuscript).

17. Methods - costs:

It is unclear what costs are being included in the annual post-hospitalisation costs. For those with mRS 3-5, they are functionally dependent and likely to have substantial care needs. In the model it includes an annual cost equivalent to US€1,746 for this cohort. Is this the cost the accrues to the payer, and the patient and their family then pay out-of-pocket for the rest? It seems an incredibly low cost to care for a functionally dependent patient for 12 months. If a very substantial cost falls on the patient and family then that must be clearly highlighted, and there should be justification for not including a societal perspective. In Ireland we were looking at annual costs of €26,604 (more than US\$30,000) for care of mRS3-5 patients.

Reply: Thanks for the reviewer's comments. the review mentioned a very important issue. As we clarified in the Costs part in the Methods (Page 7), the total costs including both out-of-pocket costs and reimbursements. The post-hospitalization cost included rehabilitation and secondary preventive costs of the patients. Our data of post-hospitalization costs was derived from the CNSR study (China National Stroke Registry) and it was credible.

However, secondary prevention and organized rehabilitation after stroke in China is highly poorer than that in the Western countries [Chin J Rehabil Theory Pract 2009;15:252-4. Chin J Convalescent Med 2009;18:472-4. (in Chinese)]. It is not surprised that post-hospitalization costs in China is much lower than that in Western countries. I guess that this situation also existed in most low- and middle-income countries. And this also suggested that it is important to perform cost-effectiveness analysis of mechanical thrombectomy in the setting of low- and middle-income countries like China.

18. Methods - health states:

How long were the Markov cycles?

Were transition probabilities static by Markov cycle?

What were the transition probabilities?

Reply: As we clarified in the Methods, the Markov cycle was 1 year (at the bottom of Page 5). As we clarified in Table 1 in the revised manuscript, transition probability of non-stroke death rate was age specific, and stroke recurrence was adjusted by relative risk of stroke recurrence per life year (Markov cycle). Other transition probabilities, including death with recurrent stroke and hazard ratio of non-stroke death for mRS 3-5, were static by Markov cycle.

19. Methods - outcome assessment:

How did you decide on the uncertainty for the utility parameters?

The figure used for mRS3-5 seems very low - looking at the various data sources used in other economic evaluations of mechanical thrombectomy suggests a figure between 0.3 and 0.4, rather than 0.21. I'd like to see better justification for the figures used.

Reply: The reviewer mentioned an important issue. As we clarified in the Methods (Page 8), utility scores of different disability states after stroke were developed in our previous study using European quality of life scale (EQ-5D) along with the Chinese preference weights in a Chinese stroke population. It is developed according to Chinese preference weights and may more appropriate to Chinese patients. The reason that the utility for mRS 3-5 was lower than that used in other countries was complicated. The potential explanation was that the organized rehabilitation and care after stroke in China is highly poorer than that in the Western countries, and stroke may induce higher economic and physiologic burden for stroke patients and the families, especially for those with moderate or severe disability. And this also suggested that it is important to perform cost-effectiveness analysis of mechanical thrombectomy in the setting of low- and middle-income countries like China.

20. Methods - sensitivity analysis:

Is the main analysis based on a deterministic model or are the summary figures derived from the probabilistic analysis?

Reply: The base-case analysis was presented in Table 2 in the revised manuscript. The results of deterministic 1-way sensitivity analysis were presented in Figure 2, and the results of probabilistic analysis were presented in Figure 3.

21. It states in the text that the costs followed log normal distributions, and yet the ranges shown in Table 2 are left- rather than right-skewed. It does not make any sense. Please explain the distribution parameters and why the skew is in the wrong direction.

Reply: Thank for the advices. We agree with the reviewer's comments. We redefined the parameters of uncertainty of costs according to the distribution of original data in the model and updated the data throughout in the revised manuscript (see Table 1 in the revised manuscript).

22. Results:

Due to some of the issues in how the input parameters have been defined and queries over the applicability of the cost data, I cannot comment on the results in detail, although I am concerned at the use of a 30 year time horizon. Ideally the tornado plot would be for the 6 year time horizon.

Reply: We thank for the reviewer's comments. The issues of definition of input parameters and applicability of cost data were clarified in the reply of corresponding comments. As we clarified in the reviewer's comments #16, most previous published studies on cost-effectiveness analysis of mechanical thrombectomy for stroke used the life-time horizons (25-30 years) in the model. Therefore, we keep the results of 30-year time horizons to compare with the studies performed in other countries. And the 1-way sensitivity analysis in Figure 2 (tornado plot), and probabilistic analysis in Figure 3 referred to 30-year time horizon.

23. Discussion:

One issue is that patients need to get to the hospital within that 6 hour time frame for mechanical thrombectomy. Is the analysis restricted to those who can get to the hospital in that time? It would have to be - in which case you should discuss the inequity for patients who cannot get there in time - they will be discriminated against. Furthermore - will mechanical thrombectomy be provided at all hospitals that provide IV-tPA? If not, then there will be a centralisation of services and some patients will require transfer to the centre doing mechanical thrombectomy - those costs have not been included.

Reply: We thank for the reviewer's comments. The reviewer mentioned an important issue. In this analysis we restricted those who can get to the hospital within 6 hours and eligible to mechanical thrombectomy since the guideline for stroke care only recommended mechanical thrombectomy to these patients. And it is right that not all hospital that provide IV-tPA will provide mechanical thrombectomy. As we clarified in the Discussion (2nd paragraph in Page 12), the real-world implement of endovascular thrombectomy treatment may be restricted by the poor awareness of the public, poor infrastructure, inefficient system and deficiency of specialists. We agree with the reviewer that the timeliness of patients got to the hospital (within 6 hours) was another important factor. We have revised the Discussion section and added the factor of timeliness of patients got to the hospital (2nd paragraph in Page 12). And we also added a limitation that costs of transfer to the hospitals doing mechanical thrombectomy has not been included in this analysis (2nd paragraph in Page 13).

24. All or almost all of the second generation trials ceased early and were sponsored by industry - are these points not worthy of a mention? They point to potential risk of bias that should be highlighted.

Reply: Thanks for the reviewer's comments. It is an important issue for the potential risk of bias of the study. We have added this issue as a limitation of the study (2nd paragraph in Page 13).

25. The limitations needs to address the use of a limited set of the available trials and clearly explain why the other four trials were not used in evidence synthesis.

Reply: We thank for the reviewer's comments. As we clarified in the reply to the reviewer's comments #15, our study focused on second-generation thrombectomy devices and consider meta-analyses included the 5 trials in 2015 could be more closed to the clinical practice in the future. We clarified this issue and emphasized that our study focused on the second-generation thrombectomy devices in the Background section in the revised manuscript (Page 4).

26. I think you should remove reference to the results being robust as shown in sensitivity analyses. The appearance of robustness is a function of structural uncertainty and how you have defined parameter uncertainty. I am unconvinced that either has been adequately addressed.

Reply: We thank for the reviewer's comments. We agree with the reviewer and deleted the sentence of reference to the results being robust as shown in sensitivity analyses in the limitation in Discussion section in the revised manuscript (1st paragraph in Page 14).

27. Table 1:

Was there no uncertainty around the proportion patients achieving mRS0-2, mRS3-5, mRS6, and sICH? There is substantial uncertainty around the risk ratios for these outcomes, and therefore there should be uncertainty in the model about the efficacy of mechanical thrombectomy.

Reply: Thank for the advices. We added the data of the 95% confidence intervals as the uncertainty around the proportion patients achieving mRS0-2, mRS6, and sICH, and the odds ratios for these outcomes in the model and updated the data throughout in the revised manuscript (see Table 1 in the revised manuscript).

28. Table 2:

Why is there no uncertainty around the proportion of patients arriving within 4.5 hours?

Reply: Thank for the advices. We added the data of the 95% confidence intervals as the uncertainty around the proportion patients arriving within 4.5 hours in the model and updated the data throughout in the revised manuscript (see Table 1 in the revised manuscript).

29. The ranges for the uncertainty on utility inputs for mRS0-2, mRS3-5, and sICH are specified incorrectly - they all skew in the wrong direction. A beta distribution with a mean greater than 0.5 will be either symmetrical or left skewed (and a beta with mean <0.5 will be symmetrical or right skewed). So the mRS0-2 range should be something like 0.60 to 0.89, not 0.60 to 0.96. How were these parameters specified?

Reply: Thank for the advices. We agree with the reviewer's comments. We redefined the uncertainty of utilities according to the distribution of original data in the model and updated the data throughout in the revised manuscript (see Table 1 in the revised manuscript).

30. Given that equal discounting is used for costs and benefits in the main analysis, why was equal discounting not applied in the sensitivity analysis (ie same lower and upper bound applied to both simultaneously)? Please justify.

Reply: We applied the different discount rate of costs and outcomes in the sensitivity analysis under the assumption that subjects may have different feeling for discounting of cost and outcome. In recent years, there is a rapid development of economics in China with an average increase rate of 7%-8% in GDP per year. Therefore, we considered a higher variance of discount rate of costs in the sensitivity analysis and set the upper limit of discount rate of cost to 8%. This may be important for the Chinese. For the discount rate of outcome, subjects' feeling may be more stable.

VERSION 2 – REVIEW

REVIEWER	Joyce Balami University of Oxford United Kingdom Involved with a multi-centre service evaluation project for mechanical thrombectomy in routine clinical practice.
REVIEW RETURNED	16-Dec-2017

GENERAL COMMENTS	Overall, the authors have addressed most of the comments and suggestions from the reviewers. However, the manuscript will benefit from proofreading for grammatical errors before publication. Minor comments: 1) Consistency with spellings for example-Hospitalization should be hospitalisation 2) The manuscript still has grammatical errors and definitely in need of grammatical improvement. The statement on Pg 38 –“ This study did not involve human subjects and therefore was exempt from institutional review board approval” is confusing, surely, this is not an animal study. I presume institutional review board (IRB) approval was not required because the authors used published data and anonymised clinical data of stroke patients. For example, the sources of human clinical data include EAST, CNSR and HERMES data.
---

REVIEWER	Conor Teljeur Trinity College Dublin, Ireland
REVIEW RETURNED	01-Dec-2017

GENERAL COMMENTS	I am pleased to see that the authors have made changes to the manuscript to reflect the reviewer comments. Some of the changes raise questions of their own, and some of the original comments have not been adequately addressed. Major revisions: As I mentioned in my original review, the paper uses clinical efficacy from an individual patient meta-analysis of 5 trials of second generation devices. I have no issue with the exclusion of the 3 first generation device trials, as it is widely accepted that those trials do not reflect the efficacy of the new devices. My issue is with the omission of data from 4 other second generation device trials (namely THERAPY, EASI, THRACE and PISTE). Inclusion of these latter trials has an impact on the estimate of clinical effectiveness. Including all the relevant evidence will reduce the OR of functional independence from 2.35 to about 2.04, which is quite a difference. I query the use of an annual Markov cycle when the probability of transition between states is measurable on a monthly scale. I'm concerned about the lack of half cycle correction as if you only allow transition at the end of a year when it may easily have occurred at any point in that year, you are biasing the costs. Furthermore, how did you deal with the discounting as you start with a three month segment, so each one year cycle will actually have 9 months in one year and 3 in the next (requiring a different discount value)? The bounds on the cost data have been revised in the Table to reflect a query on the skew of the confidence intervals. However, the bounds are incredibly wide - so for example the mean cost of annual post-hospitalisation costs for mRS0-2 is 7,385 but the bounds go from 556 to 17,801 - anything from a thirteenth of the price to 2.4 times the price. For a start this still seems a distribution skewed in the wrong direction [what are the parameters used in defining the distribution?] Either way, it strikes me that you've taken the range of values observed for individual patients in the Chinese Stroke Registry.
--

	The values used in the model should reflect the mean across patients - so ideally you would have the individual level data and calculate the bounds based on bootstrap or similar. While the mean annual cost for an individual patient could be as low as 556, it seems highly unlikely that the average across all patients with mRS0-2 could be that low. Given the issues with the cost data - I would like to see a table (it could be an online appendix) that describes the parameter values (i.e. mean and standard deviation for normal and log normal distributions, alpha and beta for beta distributions) for all distributions used in the model. Minor revisions: I am pleased to see that the confidence bounds have been added for the clinical efficacy parameters. While I appreciate that you cannot include the model itself in the paper, I have a question about how the efficacy is integrated into the model. What is presented are the proportions achieving mRS0-2 and mRS6 at 90 days in the control group. The proportions looks somewhat different when you include all 9 second generation studies (for example, 26.5% mRS0-2 goes up to 31.3%). The proportions presented are based on a fixed effect analysis when the I2 is over 70%, but I can other reasons for why you might do that but only if the OR used subsequently is based on the random effects model. You have separate odds ratios for mRS0-2 and mRS6. Superficially I might expect that you had independent distributions for the two odds ratios and applied them to the proportions in the control group to derive the proportions in the intervention group. To what extent did the modelled proportions of mRS0-2 and mRS6 in the intervention group reflect the values of a meta-analysis of proportions based on the trial data? The text should state clearly whether the efficacy data came from a random effects or fixed effect meta-analysis.
--	--

VERSION 2 – AUTHOR RESPONSE

Reviewer #1:

Overall, the authors have addressed most of the comments and suggestions from the reviewers. However, the manuscript will benefit from proofreading for grammatical errors before publication.

Minor comments:

2. 1) Consistency with spellings for example-Hospitalization should be hospitalisation

Reply: Thanks. We have revised the manuscript throughout and changed “hospitalization” to “hospitalisation”.

3. 2) The manuscript still has grammatical errors and definitely in need of grammatical improvement.

Reply: Thanks. The revised manuscript has been edited by a native English speaker. We have revised the manuscript throughout and corrected the grammatical errors.

4. The statement on Pg 38 –“ This study did not involve human subjects and therefore was exempt from institutional review board approval” is confusing, surely, this is not an animal study. I presume institutional review board (IRB) approval was not required because the authors used published data and anonymised clinical data of stroke patients. For example, the sources of human clinical data include EAST, CNSR and HERMES data.

Reply: We thank you for pointing this out. The reviewer is right. We changed this sentence into “The present study used published data and anonymised clinical data of patients from databases and therefore was exempt from institutional review board approval.” in the revised manuscript (1st paragraph in Page 6).

Reviewer: 3

I am pleased to see that the authors have made changes to the manuscript to reflect the reviewer comments. Some of the changes raise questions of their own, and some of the original comments have not been adequately addressed.

Major revisions:

5. As I mentioned in my original review, the paper uses clinical efficacy from an individual patient meta-analysis of 5 trials of second generation devices. I have no issue with the exclusion of the 3 first generation device trials, as it is widely accepted that those trials do not reflect the efficacy of the new devices. My issue is with the omission of data from 4 other second generation device trials (namely THERAPY, EASI, THRACE and PISTE). Inclusion of these latter trials has an impact on the estimate of clinical effectiveness. Including all the relevant evidence will reduce the OR of functional independence from 2.35 to about 2.04, which is quite a difference.

Reply: Thanks very much. We agree with the reviewer’s comments. We used the data of efficacy based on a meta-analysis of the 9 trials including the THERAPY, EASI, THRACE and PISTE trial. We also updated the proportion of patients received mechanical thrombectomy treatment in Table 1 according to data from the 9 trials. We clarified the issue in the Method (2nd paragraph in Page 6) and updated the data of the manuscript throughout in the revised manuscript.

6. I query the use of an annual Markov cycle when the probability of transition between states is measurable on a monthly scale. I'm concerned about the lack of half cycle correction as if you only allow transition at the end of a year when it may easily have occurred at any point in that year, you are biasing the costs. Furthermore, how did you deal with the discounting as you start with a three month segment, so each one year cycle will actually have 9 months in one year and 3 in the next (requiring a different discount value)?

Reply: Thank you very much. The reviewer mentioned an important issue. We did perform the half cycle correction for years spent in corresponding states which were then used to calculate the QALY and costs. Probability of transition between states (such as recurrent rate of stroke per patient year, annual post-hospitalization costs) were measured on a yearly scale in this study. For the first year, we started with a three months segment and 9 months segment next. We used a 1/4 discount for the first 3 months segment and 3/4 discount for the 9 months segment when we calculated the annual transition probability and costs. The discount factor for the first 3 months in the first year was $0.993 = 1/(1+0.03)^{(1/4)}$. We clarified this issue in the Method in the revised manuscript (1st paragraph in Page 6).

7. The bounds on the cost data have been revised in the Table to reflect a query on the skew of the confidence intervals. However, the bounds are incredibly wide - so for example the mean cost of annual post-hospitalisation costs for mRS0-2 is 7,385 but the bounds go from 556 to 17,801 - anything from a thirteenth of the price to 2.4 times the price. For a start this still seems a distribution skewed in the wrong direction [what are the parameters used in defining the distribution?] Either way, it strikes me that you've taken the range of values observed for individual patients in the Chinese Stroke Registry. The values used in the model should reflect the mean across patients - so ideally you would have the individual level data and calculate the bounds based on bootstrap or similar. While the mean annual cost for an individual patient could be as low as 556, it seems highly unlikely that the average across all patients with mRS0-2 could be that low.

Given the issues with the cost data - I would like to see a table (it could be an online appendix) that describes the parameter values (i.e. mean and standard deviation for normal and log normal distributions, alpha and beta for beta distributions) for all distributions used in the model.

Reply: We thank you for pointing this out. We agree with the reviewer that the values in the model should reflect the mean across patients instead of the range of values observed for individual patients. We have changed the bounds of costs to the confidence intervals of costs based on bootstrapping method (Table 1). We added a supplemental table describing the parameter values for all distributions used in the model (online supplementary table 2).

Minor revisions:

8. I am pleased to see that the confidence bounds have been added for the clinical efficacy parameters. While I appreciate that you cannot include the model itself in the paper, I have a question about how the efficacy is integrated into the model. What is presented are the proportions achieving mRS0-2 and mRS6 at 90 days in the control group. The proportions look somewhat different when you include all 9 second generation studies (for example, 26.5% mRS0-2 goes up to 31.3%). The proportions presented are based on a fixed effect analysis when the I2 is over 70%, but I can think of other reasons for why you might do that but only if the OR used subsequently is based on the random effects model. You have separate odds ratios for mRS0-2 and mRS6. Superficially I might expect that you had independent distributions for the two odds ratios and applied them to the proportions in the control group to derive the proportions in the intervention group. To what extent did the modelled proportions of mRS0-2 and mRS6 in the intervention group reflect the values of a meta-analysis of proportions based on the trial data?

Reply: Thanks very much. In the revised manuscript, we estimated the proportion of patients with mRS 0-2 and mRS 6 in the control group respectively using a meta-analysis of the 9 trials based on the random effect model (all I2 >50%). We estimated the odds ratios for mRS 0-2 and mRS 6 using a meta-analysis of the 9 trials based on the fixed effect model (all I2 <50%). The proportions in the intervention groups were then calculated based on proportions in the control group and the odds ratios for the outcomes, using the formula $p_2 = (OR * p_1) / (1 + (OR - 1) * p_1)$. We clarified this issue in the Methods in the revised manuscript (1st Paragraph in Page 7). We also checked the modelled proportions of mRS 0-2 and mRS 6 in the intervention group and proportions from meta-analysis of the trial data based on the random effect model, and found that they were very close to each other (49.6% vs. 49.9% for proportion of mRS 0-2, and 15.0% vs. 14.8% for proportion of mRS 6).

We believed the model input data of proportion of patients with the efficacy data (odds ratios) may let the readers visually understand and apply the model from the perspective of clinical application (ie. the reader can easily understand the influence of efficacy of the intervention on the results of the model). Therefore, we used data of proportion of patients with mRS 0-2, mRS 6 in the control group and the efficacy of the intervention group (odds ratios) instead of data of proportions of patients in each group as the model input data.

9. The text should state clearly whether the efficacy data came from a random effects or fixed effect meta-analysis.

Reply: Thanks. We estimated the proportion of patients with mRS 0-2 and mRS 6 in the control group using a meta-analysis based on the random effect model (all I2 >50%), and the odds ratios for mRS 0-2 and mRS 6 using a meta-analysis based on the fixed effect model (all I2 <50%). We clarified this issue in the Methods in the revised manuscript (2nd paragraph in Page 6).

VERSION 3 – REVIEW

REVIEWER	Conor Teljeur Trinity College Dublin, Ireland
REVIEW RETURNED	10-Jan-2018

GENERAL COMMENTS	I am happy to recommend the paper for publication on foot of the revisions made. I'd like to take the opportunity to commend the authors for engaging so fully in revising the paper, and being transparent about their methodology. I suggest that the editors review the paper briefly for grammar, as there are a few mistakes in the paper (e.g., "did not survived", "conducted once at a time", "worse unfavourable scenarios").
--